# Crystal structures of MHC class I complexes reveal the elusive intermediate conformations explored during peptide editing

Lenong Li [1], Xubiao Peng [2], Mansoor Batliwala[1] & Marlene Bouvier [1] ✉

Studies have suggested that MHC class I (MHC I) molecules fluctuate rapidly between numerous conformational states and these motions support peptide sampling. To date, MHC I intermediates are largely uncharacterized experimentally and remain elusive. Here, we present x-ray crystal structures of HLA-B8 loaded with 20mer peptides that show pronounced distortions at the N-terminus of the groove. Long stretches of N-terminal amino acid residues are missing in the electron density maps creating an open-ended groove. Our structures also reveal highly unusual features in MHC I-peptide interaction at the N-terminus of the groove. Molecular dynamics simulations indicate that the complexes have varying degrees of conformational flexibility in a manner consistent with the structures. We suggest that our structures have captured the remarkable molecular dynamics of MHC I-peptide interaction. The visualization of peptide-dependent conformational motions in MHC I is a major step forward in our conceptual understanding of dynamics in high-affinity peptide selection.

Antigen presentation by major histocompatibility class I (MHC I) molecules is central to adaptive immunity. MHC I molecules bind peptides and present them at the cell surface to specific receptors on CD8+ T cells. This surveillance alerts the immune system to the presence of virally infected and transformed cells. MHC I molecules bind peptides in a groove that is lined with discrete pockets (A to F)[1]. The stability of MHC class I molecules is highly dependent on interaction with a bound peptide ligand[2–5]. As such, within the endoplasmic reticulum (ER), there is an elaborate network of specialized proteins, referred to as the peptide-loading complex (PLC)[6], that ensures MHC I molecules are loaded with high-affinity peptides prior to their transport to the cell surface. Studies of the mechanism by which high-affinity peptides become ligands of MHC I molecules have highlighted that the F pocket at the C-terminal end of the groove is a critical region of conformational sensing[7–15]. Indeed, the PLC proteins tapasin, ERp57, and calreticulin are spatially organized on MHC I using binding sites at the C-terminus of the groove[16].

Molecular dynamic (MD) studies have suggested that the MHC I groove fluctuates rapidly between conformations, and it is the molecular features of these intermediate states that are recognized by the specialized proteins, particularly tapasin[7–15]. The binding of tapasin to immature MHC I molecules induces a widening of the groove which promotes the dissociation of non-optimally bound peptides[17–19]. Ultimately, under the action of tapasin, MHC I peptide repertoires are enriched in highly stabilizing peptides ensuring long-lived antigen presentation at the cell surface. Our understanding of the molecular mechanism of tapasin has benefited greatly from studies of TAPBPR[20–24], a tapasin homolog that works independently of the PLC. Although there is convincing evidence that dynamics in MHC I groove play a fundamental role in mechanisms of peptide selection and exchange[25], the molecular features of MHC I-peptide intermediates that are formed during these processes remain elusive. Except for some information on the peptide-receptive form of MHC I that revealed minor conformational adjustments in MHC I residues[26–29], the

[1]Department of Microbiology and Immunology, University of Illinois, Chicago, IL 60612, USA. [2]Center for Quantum Technology Research and Key Laboratory of Advanced Optoelectronic Quantum Architecture and Measurements, School of Physics, Beijing Institute of Technology, Beijing 100081, China. ✉e-mail: mbouvier@uic.edu

intermediate conformations available to MHC I-peptide complexes are undefined molecularly. Furthermore, it is not well understood if the region around the A and B pockets, at the N-terminus of the groove, represents another site of conformational sensing in MHC I.

In a previous study, we characterized crystallographically the presentation by HLA-B8E76C of long peptides based on the sequence (RA)$_n$AAKKKYCL ($n$ = 2 to 6)[30]. We showed that these peptides adopt canonical conformations in the groove with their extension residues (RA)$_n$ overhanging at the N-terminus of the groove. The structures also showed that the side-chain methyl group of position 1 (P1) Ala was rotated and occupied the narrow cavity that normally binds the peptide N-terminal amino group. This cavity, lined by MHC I residues Tyr7 and Tyr171[1], is critical for the A pocket structure. As such, these structures provide a unique system for evaluating experimentally the potential of the A pocket to undergo peptide-dependent conformational changes. As such, we substituted P1 Ala in (RA)$_6$AAKKKYCL 20mer with the large residue Phe generating (RA)$_6$FAKKKYCL, followed by the substitution of P2 Ala with the bulky residue Val to generate (RA)$_6$FVKKKYCL.

Here, we present the x-ray crystal structures of HLA-B8 loaded with these two peptides that reveal significant distortions and highly unusual features of MHC I-peptide interaction in the A and B pockets. The structures provide a visualization of the remarkable ability of the N-terminus of the MHC I groove to adjust conformationally in response to bound peptides, especially P1 and P2 residues. We extended these structural analyses with MD simulations that provided information about other interesting aspects of MHC I-peptide

interaction and motions that are consistent with the structures. Overall, our results describe alternate conformations available to MHC I-peptide complexes and provide a more precise understanding of how molecular dynamics support the peptide-selector function of MHC I.

## Results

### Strategically designed peptides

We designed two HLA-B8-restricted 20mer peptides based on the sequence of HIV-1 Gag epitope GGKKKYKL[31] in which P1 was substituted with Phe, P2 with either Ala or Val and P7 with Cys. The resulting FAKKKYCL and FVKKKYCL peptides were N-terminally extended with twelve residues (RA)$_6$ generating (RA)$_6$FAKKKYCL and (RA)$_6$FVKKKYCL. The P7 Cys was introduced to form a disulfide bond with Cys76 in HLA-B8, after mutating Glu76, to prevent the dissociation of long peptides from the groove[30,32]. The 8mer FAKKKYCL and FVKKKYCL control peptides were also synthesized. The reconstitution of HLA-B8E76C-peptide complexes was carried out in vitro as we described previously[30,32].

### Unconventional peptide binding and presentation

We determined the x-ray crystal structures of HLA-B8E76C loaded with the 20mer and 8mer peptides to be high-resolution (Supplementary Table 1). The structures show that FA (cyan) and FV (yellow) 20mers adopt elongated conformations in which the core residues P1 to P8 are bound inside the groove and the extension residues P-1 Ala (one position N-terminal to P1) and P-2 Arg (visible only for FV 20mer) protrude out of the groove (Fig. 1a, upper panel). The peptide

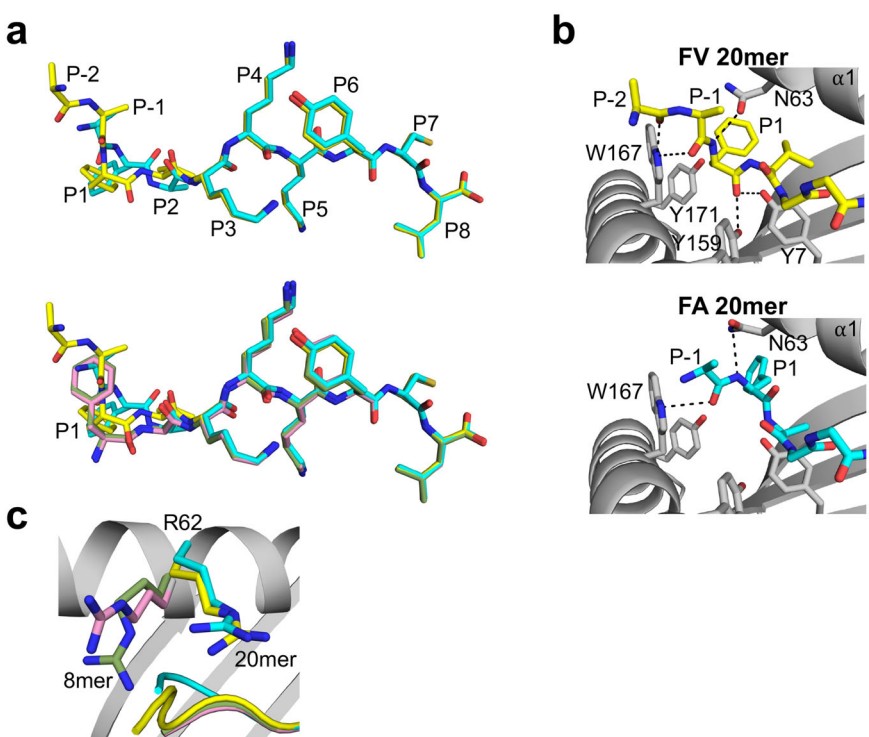

**Fig. 1 | Unusual binding and presentation of (RA)$_6$FAKKKYCL and (RA)$_6$FVKKKYCL. a** Top panel, superimposition of bound (RA)$_6$FAKKKYCL (cyan) and (RA)$_6$FVKKKYCL (yellow) 20mer peptides. The backbone and side chain conformations of the peptides overlap between P3 and P8 but differ at P1, P2, and P-1 (P-2 was visible only in 20mer FV). Bottom panel, superimposition of bound 20mer peptides with 8mer FAKKKYCL (pink) and FVKKKYCL (green) control peptides. The four peptides overlap between P3 and P8 and are most divergent at P1. **b** Interactions in the A pocket show that the main-chain nitrogen of P1 Phe FV

20mer (top panel) and FA 20mer (bottom panel) has rotated and formed a hydrogen bond with Asn63 (black dashed lines). The main-chain carbonyl oxygen in FV 20mer hydrogen bonds with Tyr159 and Tyr7, while the same atom in FA 20mer has undergone a very unusual rotation toward the α1-helix and forms no interaction with MHC I residues. In both panels, extension residues protrude out of the groove. **c** In the 20mer structures, the side chains of Arg62 have moved out of the canonical positions seen in 8mer structures, which opens the A pocket and allows the extension residues (RA)$_6$ to exit out.

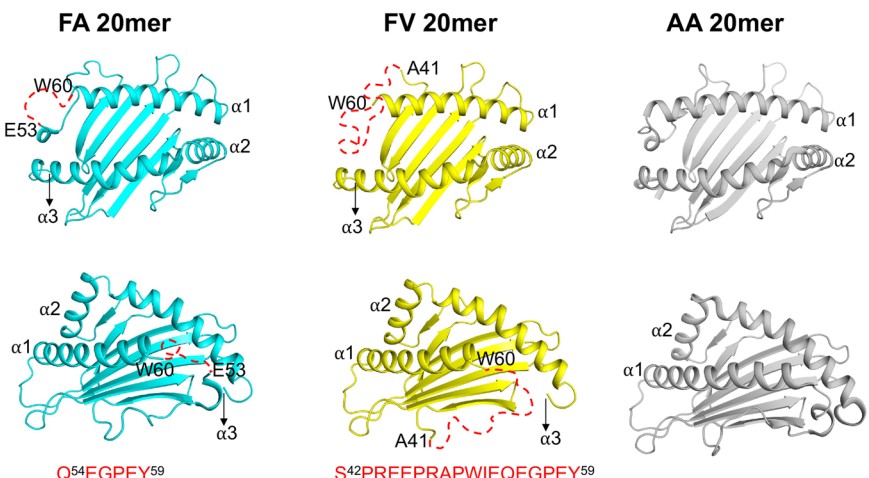

**Fig. 2 | Top and side views of HLA-B8E76C groove in FA, FV, and AA 20mer structures.** The figure shows peptide-induced structural distortions at the N-terminus of the groove. Residues Gln54 to Tyr59 (6 residues) and Ser42 to Tyr59 (18 residues) are not visible (shown as red dashed lines) in the FA and FV 20mer structures, respectively. This is in marked contrast to AA 20mer structure[30].

backbones and side chains adopt nearly identical positions between P3 and P8, but clear differences are seen at P-1, P1, and P2. The Cα-atom positions have a 2.84-Å shift at P1 and 0.96-Å shift at P2. Comparisons of FA and FV 20mers with their corresponding FA (pink) and FV (green) 8mers (Fig. 1a, lower panel) show that 8mer peptides adopt nearly identical bound conformations and that P1 is the most divergent position in both 20/8mer pairs, with shifts in Cα-atom of 3.13-Å for FAs and 1.42-Å for FVs. Finally, the electron density was clear over the entire length of the peptides, including the backbone and methyl side chain of P-1 Ala and the backbone and part of the aliphatic side chain of P-2 Arg (FV 20mer only) (Supplementary Fig. 1).

A close examination of how the FV 20mer peptide binds in the A pocket (Fig. 1b, upper panel) shows that the main-chain nitrogen of P1 Phe is rotated in a position that is normally occupied by a canonical P1 side chain, as seen in FA and FV 8mers (Fig. 1a, lower panel). In this configuration, the P1 main-chain nitrogen forms a hydrogen bond with Asn63, and the extension residues P-1 Ala and P-2 Arg protrude out of the A pocket. Interestingly, the main-chain carbonyl oxygens of P-1 and P-2 residues form hydrogen bonds with the indole nitrogen of Trp167 (Fig. 1b, upper panel), which likely stabilizes the peptide backbone as it exits out of the groove. The structure also shows that the bulky P1 Phe side chain cannot occupy the canonical position of the N-terminal amino group, i.e., the cavity formed by Tyr7 and Tyr171[1], and the phenyl ring points toward the α1-helix. Finally, the main-chain carbonyl oxygen of P1 Phe forms hydrogen bonds with Tyr7 and Tyr159 (Fig. 1b, upper panel). A similar rotation within the A pocket was also seen in the FA 20mer structure (Fig. 1b, lower panel), with P1 main-chain nitrogen and P-1 carbonyl oxygen forming hydrogen bonds to Asn63 and Trp167, respectively. In marked contrast to FV 20mer, however, the main-chain carbonyl group of P1 Phe is rotated toward the α1-helix, a highly unusual configuration, and surprisingly does not engage with any MHC I residues. Taken together, although P1 Phe residues of 20mer peptides have undergone a similar main-chain nitrogen rotation in the A pocket, there are clear differences in the binding mode of each peptide. Finally, a comparison of Arg62 in FA and FV 20mer structures relative to the 8mer structures (Fig. 1c) shows that the Arg side chains swing out of their canonical positions which creates an opening for (RA)₆ residues to exit out of the groove. A similar role of residue 62 in opening the A pocket was observed in our structure of $(RA)_6$AAKKKYCL 20mer peptide bound to HLA-B8E76C[30].

## Peptide-induced distortions at the N-terminus of the groove

The binding modes of FA and FV 20mers described in Fig. 1 are accompanied by significant structural changes in the MHC I groove. Figure 2 shows top and side views of the MHC I groove of FA and FV 20mer structures in which long stretches of N-terminal amino acid residues (shown by dashed red lines) in the α1-helix and loop connecting the α1-helix to β-strand of the floor were missing in the electron density maps. Specifically, the FA 20mer structure lacks 6 residues from Gln54 to Tyr59, and FV 20mer structure lacks as many as 18 residues from Ser42 to Tyr59 - electron densities at an acceptable 1σ threshold were not visible for these residues in our structures. Consequently, the A pocket is unstructured and widely open-ended (Fig. 2 and Supplementary Fig. 2). In contrast, our previously determined AA 20mer structure showed that the N-terminus of the groove has a canonical structure (Fig. 2)[30]. The FA and FV 20mer structures thus provide direct evidence that the N-terminus of the groove has the potential to undergo significant peptide-induced conformational distortions, highlighting its remarkable inherent plasticity. Other than these differences, minor changes in the groove were detected between the FA and FV 20mer structures (r.m.s. deviation of 0.13 Å). It is interesting that FA and FV 20mer peptides differ only by the nature of residues at P1 and P2 relative to our previous AA 20mer peptide[30]. Because all N-terminal MHC I residues were visible in the AA 20mer structure (Fig. 2), but not in the FA and FV 20mer structures, this strongly suggests that P1 and P2 residues play critical roles in the conformational maturation of the groove.

## Unusual MHC I-peptide interaction at the N-terminus of the groove

To understand the role of peptide P1 residue in modulating interactions with MHC I residues at the N-terminus of the groove, we analyzed the FA and FV 20mer structures in the context of both the AA 20mer[30] and FA 8mer structures (Fig. 3). The structure of AA 20mer shows that the small P1 Ala methyl side chain forms a hydrophobic interaction with conserved residue Tyr59 (Fig. 3a, left panel). In contrast, the large P1 Phe side chains of FA and FV 20mers sterically clash with Tyr59 (Fig. 3a, left panel), and consequently, residues Gln54 to Tyr59 become disorganized and are not visible in the FA and FV 20mer structures while these residues are visible in AA 20mer structure. Interestingly, the structure of FA 8mer (Fig. 3a, right panel) shows that the P1 side-chain phenyl ring occupies a canonical position and forms hydrophobic interactions with Tyr59, and that residues Gln54 to Tyr59 are visible. Similar observations were made relative to the FV 8mer

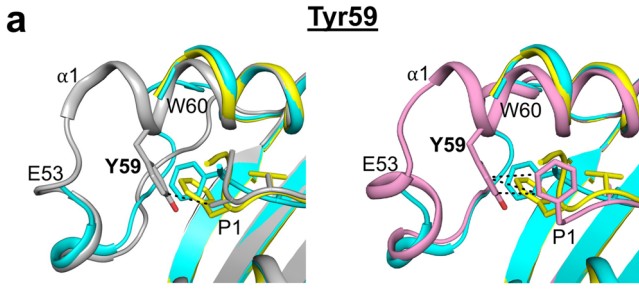

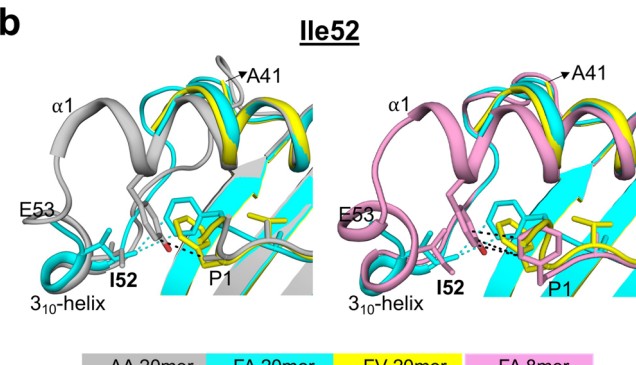

| AA 20mer | FA 20mer | FV 20mer | FA 8mer |

**Fig. 3 | Key roles of N-terminal MHC I residues Tyr59 and Ile52.**
**a** Superimposition of FA and FV 20mer structures with AA 20mer (left panel) and FA 8mer (right panel) structures. The P1 Ala side chain of AA 20mer and P1 Phe side chain of FA 8mer occupy canonical positions and interact with Tyr59 (black dashed lines), while the bulky P1 Phe side chains of FA and FV 20mers clash with Tyr59 causing conformational disorders between Gln54 to Tyr59 (left and right panels). **b** Same superimpositions as in panel (**a**). The P1 Phe side chain of FA 20mer interacts with Ile52 of the $3_{10}$-helix (cyan dashed lines) which stabilizes Ser42 to Tyr59 (left and right panels), while a similar interaction involving Ile52 is not possible for FV 20mer resulting in conformational disorders between Ser42 to Tyr59 (left and right panels). The P1 Ala side chain of AA 20mer (left panel) and P1 Phe side chain of FA 8mer (right panel) have no interaction with Ile52.

structure (Supplementary Fig. 3). Taken together, our structures indicate that the N-terminal region formed by Gln54 to Tyr59, of which residue Tyr59 is critical, has remarkable conformational flexibility enabling the groove to adapt to different sizes and orientations of peptide P1 side chains.

There is another unusual feature at the N-terminus of the groove in FA 20mer structure (Fig. 3b). In this structure (Fig. 3b, left panel), the P1 side-chain phenyl ring forms highly unusual hydrophobic interactions with the highly conserved Ile52 of the $3_{10}$-helix (a conserved element comprising residues 50 to 55). In the FV 20mer structure, however, the P1 phenyl ring is oriented differently (Fig. 3b, left panel) and, as such, it cannot engage with Ile52 thus causing residues Ser42 to Glu53 to become disordered. In the FA20mer structure, residues Ser42 to Glu53 were clearly visible due to P1 phenyl ring engagement with Ile52. Notably, interaction between a bound peptide and Ile52, or other residues of the $3_{10}$-helix, have not been reported before and are also not seen in FA and FV 8mer structures (Fig. 3b, right panel, and Supplementary Fig. 3). Taken together, the FA and FV 20mer structures show that there is a direct molecular interplay involving peptide P1 residue and N-terminal residues Tyr59 and Ile52 that influences structural integrity at the N-terminus of the groove (see also below).

## Cross-talks between peptide P1 and P2 and MHC I residues Ile52 and Tyr59

Results in Fig. 3 raise another important question: why is the P1 Phe side chain oriented differently in FA 20mer versus FV 20mer

structures? This question is important given that this difference in orientation significantly affected structural integrity between Ser42 to Glu53 in the FV 20mer structure. To address this, we examined peptide P2 residues, which is Ala in FA 20mer and Val in FV 20mer (Fig. 4a). In the FA 20mer structure (Fig. 4a, left panel), the small side-chain methyl group of P2 Ala allows the P1 main-chain carbonyl group to undergo an unusual rotation toward the α1-helix, which has the effect of orienting the P1 phenyl ring close to Ile52. In the FV 20mer structure (Fig. 4a, right panel), however, because P2 carries a larger Val side chain, the P1 main-chain carbonyl group cannot similarly rotate toward the α1-helix and consequently the P1 phenyl ring is positioned further away from Ile52. Consequently, different networks of interactions involving peptide P1 and P2 residues and N-terminal MHC I residues were overall established in FA and FV 20mer structures (Fig. 4a). Using a thermal denaturation assay, we determined similar melting temperatures (Tm) for FA and FV 20mers, 65.8 °C and 67.5 °C, respectively. In contrast, the canonical structures of FA and FV 8mers showed very similar networks of interactions (Fig. 4b) and exhibited identical Tm values, 71.1 °C for FA 8mer and 71.4 °C for FV 8mer. Finally, it is worth noting that there are two water molecules in pocket A of the 8mer structures (Fig. 4b). These water molecules form a network of hydrogen bonds with MHC I residues in pocket A, including Tyr59, that anchor the peptide N-terminal amino group. Notably, these water molecules were missing in the 20mer structures (Fig. 4a). Analysis of structures of several MHC I-peptide complexes showed that these water molecules occupy conserved positions[33]. Thus, in addition to Tyr59, water molecules play a role in the observed conformational adaptation that generates canonical pockets A and B.

## Analysis of MHC I residues in pockets along the groove
Given the importance of the six binding pockets A to F in determining the peptide side chain specificities of HLA alleles, we compared the side chain orientations of MHC I residues in pockets A and B of FA and FV 20mer structures relative to the corresponding 8mer structures (Supplementary Fig. 4). Pocket A is made up of 9 residues and typically anchors the peptide N-terminal amino group and P1 residue and closes the N-terminal end of the groove. The analysis shows that Tyr59 (conserved) and Asn63 (highly conserved) have the most divergent orientations in both FA and FV 20/8mer pairs, with some differences also seen in Tyr171 (conserved) and Trp167 (highly conserved). Pocket B is made up of 9 residues and binds the P2 peptide side chain that defines HLA binding motifs. For both the FA and FV 20/8mer pairs, all MHC I residues adopted very similar orientations, except for Asn63 at the boundary of pockets A and B. A similar analysis for pocket C showed no changes in MHC I side chain orientations (Supplementary Fig. 4), with identical conclusions made for pockets D to F. Taken together, the binding of 20mer peptides affected the configuration of MHC I residues in pocket A more significantly than those in pocket B, consistent with P1 being the most divergent position of these peptides (Fig. 1a). This analysis also highlighted a critical role for Asn63 in MHC I maturation; while Asn63 mediated several hydrophobic and hydrogen bond interactions with P1 and P2 residues in the FA and FV 20mer structures (Fig. 4a), Asn63 formed only one hydrogen bond with P2 main-chain nitrogen in FA and FV 8mer structures (Fig. 4b).

## Conformational flexibility and geometrical parameters
To further characterize HLA-B8E76C-peptide complexes and the interaction between peptide P1 residue and Tyr59 and Ile52, we analyzed thermal properties, i.e., conformational flexibility and geometrical parameters, from MD simulations at physiological temperature. The equilibration of MD simulations was validated by time-course analysis (Supplementary Figs. 5 and 6).

The flexibility of individual MHC I residues along the heavy chain is characterized by its root mean square fluctuation (RMSF) (Fig. 5a). Results show that RMSF values are clearly higher in the region

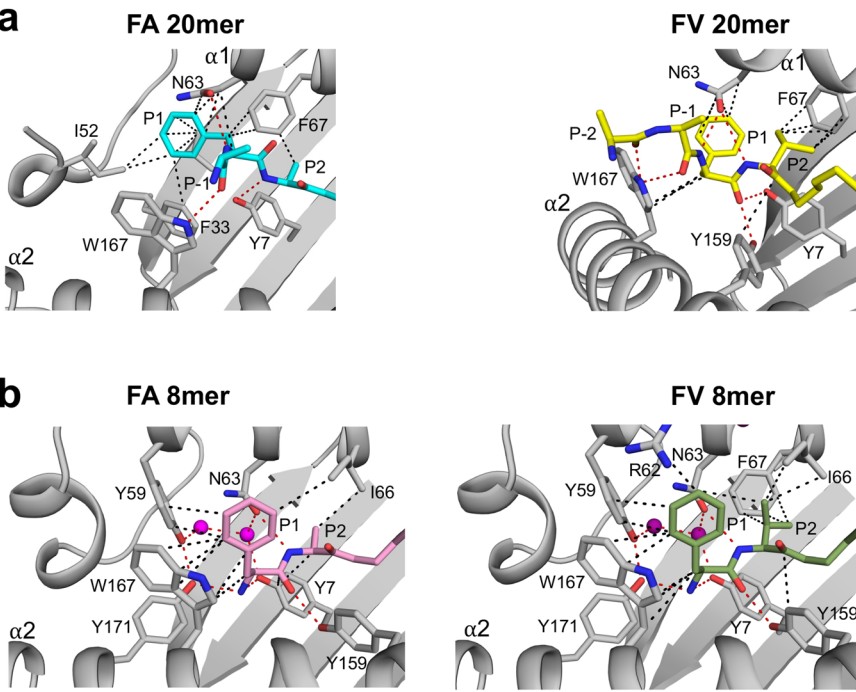

**Fig. 4 | Molecular cross-talks between occupied pockets A and B. a** P1 phenyl side chain is oriented differently in FA 20mer (left panel) relative to FV 20mer (right panel). This is due to molecular cross-talks between peptide P1 and P2 residues and Ile152 and Tyr59 (missing) (see text and Fig. 3b). Overall, the network of hydrophobic (black dashed lines) and hydrogen bond (red dashed lines) interactions in the A and B pockets are different in these two structures. **b** The network of interactions in FA 8mer (left panel) and FV 8mer (right panel) are quite similar overall in these canonical structures, which is in contrast to FA and FV 20mer structures shown in panel a. Water molecules (shown as magenta balls) are involved in a network of hydrogen bonds.

comprising residues 41 to 62 (shown in a red box) for FA and FV 20mer complexes relative to other regions, indicating that residues 41 to 62 are conformationally more flexible and thermally more unstable. A zoom-in revealed two distinct regions (Fig. 5b); residues 41 to 46 and residues 52 to 62. In the region of residues 41 to 46, all complexes have similarly high RMSF values, suggesting that conformational flexibility is independent of the bound peptide in this region. However, in the region of residues 52 to 62, the RMSF values for FA and FV 20mer complexes are much higher than those of the other complexes, indicating that the 20mer complexes are conformationally more flexible in this region.

Next, we evaluated the geometrical parameters of interaction between peptide P1 residue and Tyr59 and Ile52 to gain further insights into these complexes (see Methods). Figure 5c shows the probability distributions of inter-residue distance between P1 and Tyr59 for all five systems. Results show that the peak of the distribution is centered around 9 Å for AA 20mer while the other distributions are centered around 6 Å, suggesting a high probability of interaction between P1 and Tyr59 in all complexes with FA and FV 8/20mers when equilibrated at physiological temperature. However, because the distributions are broader for FA and FV 20mers relative to the control 8mers, it suggests that the free energy landscape of the long peptides is shallower relative to the control peptides, with the distance between P1 and Tyr59 as the reaction coordinates. Therefore, considering thermal fluctuations, the interaction between P1 and Tyr59 is more likely to break in FA and FV 20mers. We also calculated the probability distributions of relative orientation between the aromatic rings of P1 Phe and Tyr59 (Supplementary Fig. 7a). Results show that FA and FV 8mers have symmetric distributions centered around 90°, indicating that the two aromatic rings are essentially perpendicular to each other. In contrast, for FA and FV 20mers, the peaks of the distributions are positioned at 110° and 150°, respectively, showing that the aromatic rings of P1 Phe and Tyr59 are no longer perpendicular to each other. Accordingly, we conclude that the configuration of π-π interactions[34] between P1 Phe

and Tyr59 is more parallel-like in FV 20mer but changes toward T-shape-like in FA 20mer, and finally adopts the stable T-shaped structure in FA and FV 8mers. Finally, Fig. 5d clearly shows that the probability distributions of inter-residue distance between P1 and Ile52 are different among the five complexes. For FA 20mer, the distribution has a high peak centered around 5 Å, indicating that there is a stable interaction between these two residues. For FV 20mer, the distribution still peaked around 5 Å but it is much wider, indicating that interaction between P1 and Ile52 breaks more easily owing to thermal fluctuations (Fig. 5d). In contrast, the distributions for the control 8mers and AA 20mer have high peaks centered around 10 Å or above, indicating that there are no interactions between P1 and Ile52 in these complexes. To further characterize the interaction between P1 and Ile52 in FA and FV 20mers, we determined the probability distributions of angle C-H-X (X being the center of P1 Phe aromatic ring) (Supplementary Fig. 7b). Results show that the angles are mostly in the range 120° to 150°. Taken together, since the probability distributions of "distance" (Fig. 5d) and "defined angle" (Supplementary Fig. 7b) between P1 and Ile52 satisfy the geometric criterion of CH-π interaction[35], we conclude that CH-π interaction exists in both FA and FV 20mers. Finally, because the peptides are covalently bound to the MHC I groove via an engineered disulfide bond at P7, the MD simulations were repeated in conditions where the disulfide bond that tethers FA and FV 20mers is reduced (Supplementary Fig. 8). The results showed the properties of both types of systems, i.e., reduced versus non-reduced peptide disulfide-bond, are essentially unchanged and thus our main conclusions on structural adaptation in the A and B pockets remain unaffected.

**Rotation of peptide N-terminal amino group in the A pocket**
Given that the N-terminal amino groups of FA and FV 20mers adopt highly unusual configurations in the A pocket (Fig. 1b), we conducted MD simulations to assess whether the amino groups can undergo spontaneous rotations to canonical positions, i.e., point down in the A pocket. For these tests, we removed the extension (RA)₆ residues of FA

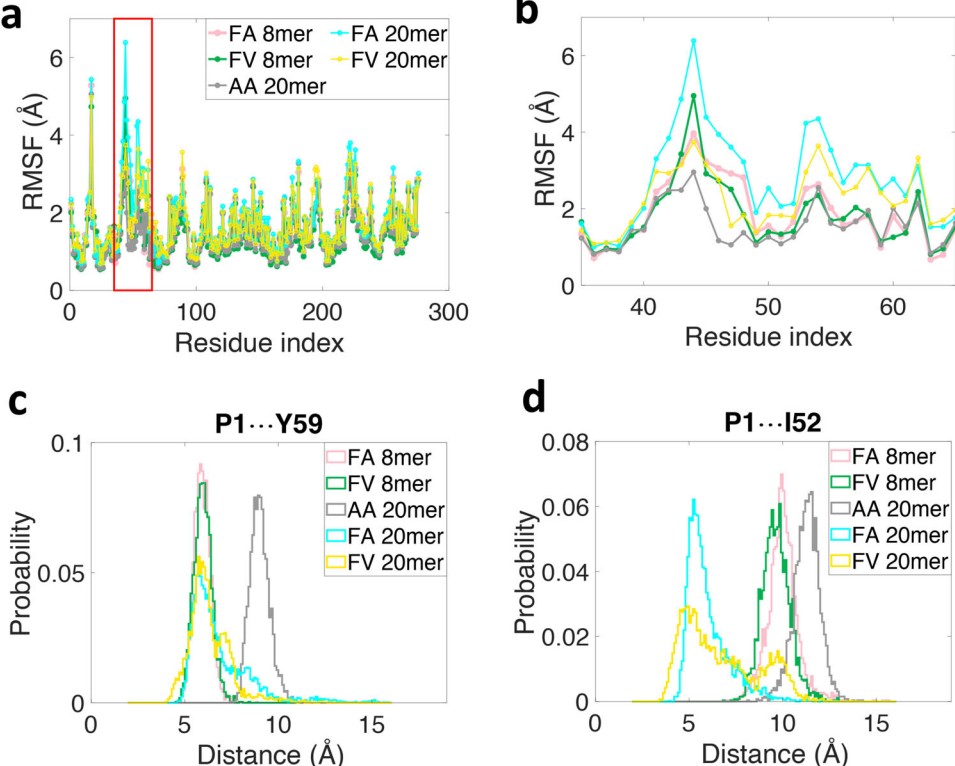

**Fig. 5 | Properties of equilibrium ensembles of HLA-B8E76C-peptide complexes. a** Root mean square fluctuations (RMSF) values of individual residues along the heavy chain of HLA-B8E76C loaded with different peptides highlighting that the highest values are in the region of residues 41 to 62 (shown in a red box). **b** A zoom-in of panel (**a**) reveals two distinct regions, residues 41 to 46 (peptide-independent) and residues 52 to 62 (peptide-dependent). **c** Probability distributions of inter-residue distance between peptide P1 and Tyr59 (P1⋯Y59) and (**d**) between peptide P1 and Ile52 (P1⋯YI52) for HLA-B8E76C loaded with different peptides (see text).

The inter-residue distances between P1 and Tyr59 are defined between the geometrical centers of the two aromatic rings for FA and FV 8/20mers, and are defined between the Cβ atom of P1 Ala and the geometrical center of the aromatic ring of Tyr59 for AA 20mer. Similarly, the inter-residue distances between P1 and Ile52 are defined between the geometrical center of the aromatic ring in P1 Phe and Cδ atom of Ile52 for FA and FV 8/20mers, and between Cβ atom in P1 Ala and Cδ atom of Ile52 for AA 20mer.

and FV 20mers in the structures, generating FA20..8mer and FV20..8mer (see Methods). The dihedral angle ω that defines the rotation of the N-terminal amino group is shown in Supplementary Fig. 9a for FV20..8mer as an example. In FA and FV 20mer structures, ω values are −110° and −73°, respectively, indicating that both N-terminal amino groups point up, while in FA and FV 8mer structures, ω values are 96° and 98°, respectively, indicating that the amino groups point down.

The time traces of the dihedral angle ω in MD simulations for FA20..8mer and FV20..8mer are shown in Supplementary Figs. 9b and c, respectively. For FA20..8mer, the results show that ω changes from about −100° to 460°/−80°/−260° for three replicas within 1 μs (note that ω values of 460°/−260° are equivalent to ω = 100°, when the periodicity of ω is taken into consideration). For FV20..8mer, ω changes from approximately −70° to −260° in all replicate simulations within 1 μs. Hence, we conclude that the N-terminal amino groups of FA20..8mer, and FV20..8mer have high probability to rotate in the A pocket and point down as seen in canonical structures. Finally, a similar analysis using non-disulfide bonded FA20..8mer and FV20..8mer led to the same conclusion on P1 rotation in the A pocket, where two out of three replicas reached the canonical position for both complexes (Supplementary Fig. 10).

### Comparison of our structures with bat MHC I and human MHC II molecules

The characterization of bat MHC I genes identified thus far, showed that many of these molecules contain a 3- or 5-amino acid insertion in their binding groove[36]. Several structures of Ptal-N*01:01 bat MHC I

molecules with a 3-amino acid insertion have been determined and revealed that the insertion creates a turn at the N-terminus of the groove (Fig. 6a)[37,38], within the critical region of Gln54 to Tyr59 (shown in red). The bat structures also revealed that in this turn, Asp59 forms salt bride interactions with Arg65 of the α1-helix (Fig. 6a)[37,38]. Such a pairing of charged residues at these positions is a highly conserved feature of bat MHC I molecules[37,38], and may alter the conformational flexibility of the groove in ways that remain to be determined.

Finally, in MHC class II (MHC II) molecules, there is evidence of peptide-induced conformational changes at the N-terminus of the groove[39–41], i.e., the region recognized by the peptide-exchange catalyst HLA-DM[42,43]. The binding of HLA-DM induces structural changes in MHC II, particularly in the $3_{10}$-helix and unstructured loop[44] (shown in dark blue in Fig. 6b). This critical region of MHC II overlaps with the region of MHC I ($3_{10}$-helix and extended region) that we identified as important for shaping the A and B pockets (shown in red in Fig. 6b). This observation lends support to the view that the N-terminus of the groove may have a role in peptide editing (see Discussion).

## Discussion

The crystal structures of peptide-filled MHC I molecules have taught us a great deal about molecular recognition of bound peptides. These structures represent the endpoint of a complex intracellular maturation process whereby MHC I molecules acquire peptides of sufficiently high affinity to ensure efficient antigen presentation. Biophysical, NMR, and in silico studies have been consistent in demonstrating that MHC I molecules explore intermediate conformational states in solution during peptide binding. Because these canonical structures are

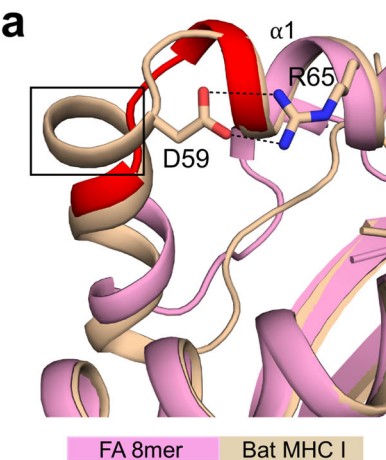

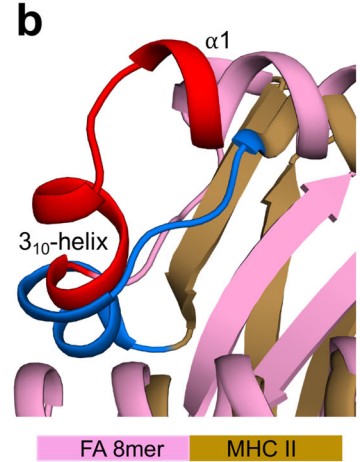

**Fig. 6 | Comparisons with the structures of bat MHC I and human MHC II molecules. a** Superimposition of FA 8mer and bat MHC I molecule (Ptal-N*01:01; PDB accession number 6J2D) structures. The bat molecule has an additional turn (highlighted by a black box) in the extended region of Gln54 to Tyr59 (highlighted in red) that we identified as disordered (see Fig. 2). In this turn, Asp69 forms salt bridge interactions (black dashed lines) with Arg65 of the α1-helix.

**b** Superimposition of FA 8mer and human MHC II molecule (HLA-DR1; PDB accession number 1DLH) structures showing that the HLA-DM susceptible region, i.e., $3_{10}$-helix and unstructured loop (highlighted in dark blue) overlaps with the $3_{10}$-helix and extended region (highlighted in red) that we identified as critical in shaping the A and B pockets (see Fig. 4).

always nearly identical, it has not been possible thus far to derive knowledge of intermediate conformations explored by MHC I molecules. This makes the notion of molecular dynamics in MHC I and of peptide-dependent motions elusive. We have determined the crystal structures of HLA-B8 loaded with 20mer peptides that reveal highly unusual features in both MHC I and peptides. We have also carried out MD simulations to gain further insights into peptide-dependent interaction in these complexes, which provided additional information about these molecules.

The FA and FV 20mer structures showed that the N-terminus of the groove has undergone significant distortions relative to FA and FV 8mer structures (Fig. 2 and Supplementary Fig. 2). These differences were peptide specific. MD simulations identified that residues 52 to 62 are most conformationally flexible and thermally unstable in FA and FV 20mer complexes relative to the other complexes (Fig. 5b), consistent with the lack of electron density for residues 54 to 59 in FA and FV 20mer structures. MD simulations also identified a conformationally flexible region comprising residues 41 to 46 that is more independent of peptides. Consistent with this, it is interesting that several deposited structures of HLA-B8 loaded with 9mer peptides lack clear electron density between residues ~41 to 49 (for example, 1M05, 3SKO, 4QRQ, and 5WMR), suggesting that some conformational fluctuations can persist at the N-terminus of the groove in canonical structures. Our structures also showed that the 20mer peptides, with only a single amino acid difference at P2 (Ala versus Val), adopted different and unusual backbone and side chain orientations at P1 and P2 but overlapped almost identically between P3 to P8 (Fig. 1a, upper panel). Interestingly, both complexes had similar thermostabilities, 65.8 °C (FA 20mer) versus 67.5 °C (FV 20mer). We suggest that the FA and FV 20mer structures, although static snapshots, represent intermediate conformations (from an ensemble) available to MHC I molecules, exemplifying the remarkable molecular dynamics of these molecules.

MHC I peptide ligands are usually 8 to 10 amino acids long and bind with their N-terminal amino group pointing down in the A pocket, as seen in the FA and FV 8mer structures (Fig. 4b). Because short linear peptides are generally unstructured in solution, it is reasonable to assume that they land as such within the immature MHC I groove. NMR and other biophysical studies showed that in the initial binding steps, incoming peptides are loosely accommodated in the groove until more specific conformational adaptions take place in both peptides and MHC I[24,25,45,46]. It is therefore plausible that when peptides of

optimal lengths are first captured by MHC I, they adopt conformations that resemble those of our 20mer peptides, i.e., with an unusual rotation of the N-terminal amino group in the A pocket (Fig. 1b), until folding proceeds and peptides adopt a canonical conformation (or not). We simulated the relaxation process of FA20..8mer and FV20..8mer using plain MD simulations and the results showed that peptide N-terminal amino groups have a high probability to rotate to a canonical position from the unusual orientations observed in our structures. The simulations also suggested that the configurations of FA and FV 20mer peptides, as seen in the structures, represent "trapped" states (see below). Finally, it is worth noting that naturally occurring HLA-B8-restricted 8 and 9mer peptides with a large residue at P1 or at P1/P2 have been reported: for example, Phe, Val, Leu, and His at P1[47–50] or Tyr/Leu, Trp/Val, Phe/Leu, and Tyr/Ile at P1/P2[47,51,52].

The FA and FV 20mer structures showed that strictly conserved Tyr59 and highly conserved Ile52 ($3_{10}$-helix) act synergistically at the N-terminus of the groove. In canonical MHC I structures, Tyr59 stabilizes the peptide N-terminal amino group together with Tyr171, and Ile52 acts as a structural support to Tyr59 and Tyr171[1]. In our 20mer structures, Tyr59 could not adopt its canonical position because of the large size of the P1 Phe side chain in the A pocket (Fig. 3a). This caused residues 54 to 59 to become disordered and created an open-ended A pocket. In the FA 20mer structure, this disordering was accompanied by Ile52 forming a highly unusual interaction with peptide P1 Phe side chain, which in turn was facilitated by the small peptide P2 Ala side chain in the adjacent B pocket (Fig. 4a). In the FV 20mer structure, however, similar molecular cross-talks between the A and B pockets were not possible because of the larger peptide P2 Val side chain in the B pocket that positioned P1 Phe side chain further away from Ile52 (Fig. 4a). As a result, a significantly longer stretch of residues became disordered in the FV 20mer structure, generating a widely open-ended A pocket. In performing MD simulations, we were able to probe other aspects of the interaction between peptide P1 and MHC I residues Tyr59 and Ile52 in the thermal equilibrium ensembles of our complexes. The simulations revealed that 1. interaction between P1 and Tyr59 is characterized by the more stable T-shaped π-π configuration in FA and FV 8mer complexes relative to FA and FV 20mer complexes (Fig. 5c and Supplementary Fig. 7a), which is consistent with residues 54-59 being ordered in the 8mer structures but disordered in the 20mer structures; and 2. CH-π interaction between P1 and Ile52 exists only in FA and FV 20mer complexes, and it is more

stable in FA 20mer than FV 20mer (Fig. 5d and Supplementary Fig. 7b). This is also consistent with the FV 20mer structure showing a more highly disordered groove and widely open-ended A pocket. Taken together, we suggest that the 20mer structures represent "trapped" states, i.e., conformations in which Tyr59 cannot adopt a "closed" position (FA and FV 20mer) and $3_{10}$-helix cannot mature into its secondary fold (FV 20mer). In other words, the formation of the A and B pockets requires that Tyr59 transitions into a "closed" position and $3_{10}$-helix (Ile52) into its canonical fold. A role for the $3_{10}$-helix in peptide-dependent MHC I folding was suggested previously[53–55]. Because Tyr59 and $3_{10}$-helix are strictly conserved elements in human HLA alleles, these transitions are expected to be universal in MHC I maturation. We also showed that solvation has a role in conformational adaptations in pockets A and B. That the 20mer structures can tolerate such high degrees of peptide-induced structural perturbations at the N-terminus of the groove is consistent with MD simulations of other groups showing that bound peptides can dissociate partially at the N-terminus of the groove while remaining anchored within the F pocket[56–59], and that these motions may have functional implication (see below). Finally, the lower thermostabilities of FA and FV 20mer complexes relative to the 8mer complexes are consistent with the "trapped" states being higher energy states, but still of sufficiently low energy to be captured in crystallography.

The conformational transitions discussed above are intricately coupled to the occupancy of the A and B pockets. This is significant when considering that the crystal structure of a peptide-free MHC I molecule (HLA-A2) showed no substantial differences at the N-terminus (or C-terminus) relative to canonical HLA-A2 structures, except for minor differences in some side chain orientations within the A pocket[30]. It is also significant because, in canonical MHC I structures, the A and B pockets usually anchor the peptide N-terminal amino group and P2 primary anchor residue, respectively, which contribute to protein stability. The FA and FV 20mer structures thus likely captured intermediate conformations that MHC I adopts transiently when "evaluating" peptide P1 and P2 residues, with the exact molecular features of these intermediates expected to be fluid based on our MD simulations.

Previous studies indicated that MD around the F pocket is a major driving force for the recognition of MHC I molecules by tapasin and TAPBRP[7–15,18,19]. Conformational flexibility at the N-terminus of the groove of MHC II molecules is also a critical determinant of HLA-DM-mediated mechanism of peptide exchange[44]. As such, our structures of FA and FV 20mers raise the intriguing question of whether the region close to the A and B pockets represents a binding surface recognized by a protein, yet to be identified, with a role in stabilizing MHC I-peptide intermediates and/or peptide editing. It is interesting that the cryo-EM structure of the PLC showed that the long P-domain of CRT chaperone is positioned atop and across the MHC I groove with its tip interacting with ERp57[16]. In this spatial organization, the P-domain could interact transiently with the partially folded region of pockets A and B, as facilitated by the inherent conformational flexibility of the P-domain and dynamic nature of the P-domain/ERp57 interaction[60,61]. More work is required to examine this idea. The possibility that ERAP1, ERAP2, and/or ERAP1/ERAP2 heterodimer play a functional role at the N-terminus of the groove is also very reasonable. We showed previously in biochemical studies that ERAP1 and ERAP1/ERAP2 can trim the protruding N-terminal amino acid residues of long peptides (>14mer), including the AA 20mer peptide, while bound to HLA-B8E76C[30,32]. Others have also shown that ERAP1 trims peptides bound to H2-K$^b$ using cell-based assays[56]. Moreover, it was demonstrated that mouse ERAAP (equivalent to ERAP1) synergizes with tapasin to edit peptide repertoires[62]. In fact, a role for MHC I molecules in antigen processing has long been suggested[63]. Our current results further support a functional role for ERAP enzymes at the N-terminus of the groove and also suggest that the enzymes are more likely engaging with intermediate forms of MHC I molecules, rather than canonical MHC I molecules[64].

In conclusion, our study provided a crystallographic and MD characterization of molecular flexibility and conformational substates in MHC I-peptide complexes. These results are important to understand the molecular mechanism by which MHC I-restricted peptide repertoires are developed in antigen presentation. Finally, our work opens avenues to examine chaperoning at the N-terminus of the groove around the A pocket.

## Methods

### Synthetic peptides
Peptides were synthesized by the solid-phase methodology (GenScript Biotech Co.) and purified by reverse-phase chromatography on a C18 HPLC column. Stock solutions of peptides in DMSO were stored at −80 °C.

### Refolding of HLA-B*0801E76C complexes
Using the crystal structure of HLA-B*0801/GGRKKYKL (Protein Data Bank (PDB) accession number 1AGB), we identified residue Glu76 to be geometrically well-positioned to form a disulfide bond with the side chain of P7 peptide residue, after mutation with a cysteine[32]. The HLA-B*0801E76C heavy chain mutant was generated using standard cloning procedures[32]. HLA-B*0801E76C complexes were reconstituted from urea-solubilized inclusion bodies of HLAB*0801E76C heavy chain (1 μM) and β$_2$-microglobulin (2 μM) with a synthetic Cys-P7 peptide (10 μM) in an oxidative refolding buffer at 4 °C[65]. After 48 h, the crude refolding mixture of HLA-B*0801E76C complexes was purified on a Superdex-200 size exclusion chromatography column by FPLC. Stock solutions of purified complexes (10–30 mg/ml) in 20 mM Tris-HCl, pH 7.5, 150 mM NaCl, were kept at −80 °C.

### Crystallization
The initial crystallization condition of HLA-B*0801E76C/(RA)$_6$ FAKK-KYCL (10 mg/ml) was identified using the Crystal Screen™ (Hampton Research, Riverside, CA) as solution #9 (0.2 M ammonium acetate, 0.1 M sodium citrate tribasic dihydrate, pH 5.6, 30% (w/v) PEG 4000) via the hanging-drop vapor diffusion method at room temperature. The initial crystals were optimized using different pH values (4.5–7.0) and PEGs (6000–10000; 10–30%). These optimized crystals were used to generate a seeding solution in solution #9. Crystals used for data collection were grown by mixing 2 μl of 10 mg/ml protein solution with 2 μl of 0.2 M ammonium acetate, 18% PEG 4000, 0.1 M sodium citrate, pH 5.7, and 0.5 μl of seeding solution. Similar crystallization conditions were used to collect data for HLA-B*0801E76C loaded with FA and FV 8mers and FV 20mer.

### Structure determination and refinement
X-ray diffraction data sets were collected at 100 K with a MAR-225 CCD detector at the LS-CAT beamlines 21-ID-G and 21-ID-F at the Advanced Photon Source, Argonne National Laboratory (Lemont, IL). Data were integrated and scaled with the HKL-2000 program package 717[66] or XDS-x86[67]. Details of data processing are presented in Supplementary Table 1. The structures of all complexes were solved by molecular replacement using Phaser 2.3.0[68] (the initial search model was HLA-B*0801E76C/R(N-Me)AAAKKKYCL (PDB code 6P2S)). Structure refinement of all models was carried out in Phenix 1.20.1 (or Refmac in CCP4 7.1.018)[69–71] and manual building with COOT 0.6.2[72]. Final refinement statistics are summarized in Supplementary Table 1. The atomic coordinates of all structures have been deposited in the PDB with the following accession codes: 8E13 (FA 8mer), 8E2Z (FA 20mer), 8E8I (FV 8mer), and 8EC5 (FV 20mer).

### Thermal denaturation assay
A thermal denaturation assay was performed using reaction mixtures consisting of 7 μl of a complex (final concentration of 2 μM), 7 μl of 10x SYPRO orange dye (5000x, Thermo Fisher Scientific, Waltham, MA)

and 7 μl of 50 mM HEPES, pH 7.2, 150 mM NaCl. Each mixture (total volume 21 μl) was analyzed in quadruplicate using an ABI ViiA7 RT-PCR instrument (Life Technologies, Inc., Carlsbad, CA). A temperature gradient from 25 to 95 °C with continuous increments of 0.06 °C/sec was used to generate the denaturation curves. The averaged denaturation curves were plotted as "fluorescence intensity" versus "temperature", and the minimum point of the first derivative of each curve provided the melting temperature.

## MD initial structures preparation

The x-ray crystal structures of HLA-B8E76C loaded with five different peptides were analyzed by MD simulations: FA and FV 8mers (this study), AA 20mer (6P2C)[30], and FA and FV 20mers (this study). The missing regions of HLA-B8E76C in the structures of FA and FV 20mers were complemented using the software UCSF Chimera 1.14[73] with the structure of AA 20mer as the template. The missing regions of the bound 20mer peptides were complemented using Modeller 9.15[74] integrated into software UCSF Chimera 1.14[73]. In the simulations, the peptide termini were neutralized to exclude artificial charge effects. For FA/FV 20mers with a reduced disulfide bond between Cys76 and peptide P7 Cys, the model HLA-B8E76C was started by breaking the disulfide bond and adding hydrogen atoms covalently bonded to the sulfur atoms of Cys76 and P7 Cys.

## MD simulations setup

Depending on the purpose, two different all-atom MD simulations were performed: replica exchange MD (REMD) and plain MD. The REMD simulations were used for the thermal properties analysis due to their efficiency in thermal equilibrated conformational ensemble sampling, while the plain MD simulations were used for simulating the rotation of the N-terminal amino group of bound FA20..8mer and FV20..8mer from their unusual up orientations. Both types of MD simulations included common settings. Simulations were performed with explicit solvent using the software package GROMACS 5.1.1[75,76]. The force field CHARMM36m[77] together with its own modified TIP3P water model[78] was also used. The protonation states of the residues were assigned using the GROMACS pdb2gmx program by default at pH 7. The LINCS algorithm was applied to constrain covalent bonds with H-atoms, and the time step in the simulation was 2 fs. The electrostatic interactions were calculated using the particle mesh Ewald (PME) method and the cutoff distance was set to 1.2 nm. The cutoff for the van der Waals interactions was also set to 1.2 nm. The protein was simulated in 0.1 M aqueous NaCl solution. After short energy minimization, an NVT simulation of 100 ps with the V-rescale temperature coupling at 310 K was performed, followed by an NPT simulation of 300 ps with the Parrinello-Rahman coupling method at a reference pressure of 1 bar. The relaxation times for the temperature coupling and pressure coupling are 0.1 ps and 2 ps, respectively. During NVT and NPT simulations, the protein backbone was restrained to its initial structure. At the end, we removed the constraints and performed production simulations at the same temperature and pressure. Coordinates were saved to disk every 20 ps. Other information on MD simulations setup is presented in Supplementary Table 2.

## REMD simulations

After a short thermal equilibration process with NVT and NPT, as described above, we set up 30 replicas with temperatures distributed from 300 K to 340 K following the webserver (https://virtualchemistry. org/remd-temperature-generator/)[79] with a swap attempt frequency of 1/ps between two neighboring temperatures. The average acceptance probability for the replica exchanges was about 30%. For FA 8mer, FV 8mer, and AA 20mer, each replica ran for 50 ns, and thus we have a simulation with a total time of up to 1.5 μs. For FA 20mer and FV 20mer with residues 54–59 not resolved in the x-ray structures, each replica

ran for 90 ns and we have a simulation with total time up to 2.7 μs. The convergence of the simulations was checked by time-course analysis on RMSF and the focused geometry parameters (Supplementary Figs. 5 and 6). The thermally equilibrated ensembles were collected from the replica at temperature 310 K from the REMD simulations, from which the flexibility of HLA-B8E76C heavy chain and interactions between peptide residue P1 and MHC I residues Tyr59 and Ile52 were analyzed.

## Plain MD simulations

Using the FA and FV 20mer structures, as generated above (see "MD Initial structures preparation"), we generated the initial configurations of FA20..8mer and FV20..8mer by deleting the extension $(RA)_6$ residues. After a short thermal equilibration process with NVT and NPT as described above, we performed the production run for 1μs to simulate the relaxation process of FA20..8mer and FV20..8mer. During the simulations, we observed rotations of the N-terminal amino groups in FA20..8mer and FV20..8mer peptides. We repeated the simulation three times for each system to ensure reproducibility.

## Interaction analysis

We identified possible interactions between peptide residue P1 and MHC I residues Tyr59 and Ile52 according to the geometrical properties, i.e., distances and relative orientations between the two residues. For interaction between P1 Phe and Tyr59, the distance was defined between the geometrical centers of the two aromatic rings, while the relative orientation was defined by the angle between the normal directions of the aromatic rings. For interaction between P1 Ala of AA 20mer and Tyr59, the distance was defined between the Cβ atom of P1 Ala and the geometrical center of the aromatic ring of Tyr59. For interaction between P1 Phe and Ile52, the distance was defined between the geometrical center of the aromatic ring in P1 Phe and the Cδ atom of Ile 52, while the angle C-H-X was defined by the atoms Cδ of Ile52, the hydrogen atom covalently bonded to Cδ and the center of Phe aromatic ring. Finally, for interaction between P1 Ala of AA 20mer and Ile52, the distance was defined between the Cβ atom in P1 Ala and the Cδ atom of Ile52.

## Analysis of peptide N-terminal amino group rotations

We probed rotations of N-terminal amino groups in pocket A by monitoring time traces of the dihedral angle ω formed by atoms Cys76:CA-P7:CA-P1:CA-P1:N (Supplementary Fig. 9a) in the plain MD simulations. When drawing time traces, we reasonably required that the rotation between two neighboring frames be less than 180°, i.e., $|\omega(t_{i+1}) - \omega(t_i)| < 180°$. Otherwise, we added/subtracted the dihedral angle $\omega(t_{i+1})$ by 360°. In the end, to eliminate thermal fluctuations and visualize the rotation pathways better, we smoothed the time traces by an average sliding window of 10 ns. VMD 1.9.2 software[80] was used to calculate all geometry parameters in "Interaction analysis" and "Analysis of peptide N-terminal amino group rotations".

## Data availability

The coordinates and structure factors generated in this study have been deposited in the PDB under accession numbers 8E13 (FA 8mer), 8E2Z (FA 20mer), 8E8I (FV 8mer), and 8EC5 (FV 20mer). Our previously solved structure of AA 20mer has PDB accession number 6P2S, other structures used in the study have codes 1DLH, 6J2D, and 1AGB. All relevant data are within the paper and its Supplementary Information. Other data are available from the corresponding author upon request.

## Code availability

Input, output, and parameter files for the MD simulations performed are supplied in Supplementary Data 1 and 2.

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

## Acknowledgements

We thank Dr. Bernard Santarsiero for help with remote x-ray data collection and discussion, and the staff at the Advanced Photon Source at Argonne National Laboratory where all x-ray data were collected. This work was supported in whole, or in part, by the National Institutes of Health (NIAID) Grants R01 AI114467 (MB), R01 AI108546 (MB), and R21 AI173863 (MB).

## Author contributions

L.L. executed all experiments including x-ray crystallography and was assisted by M.B. in refolding MHC I-peptide complexes. L.L. and M.B. designed experiments and interpreted the structures. X.P. performed M.D. simulations and interpreted the results. M.B. wrote the manuscript with contributions from X.P. and L.L.

## Competing interests

The authors declare no competing interests.
