## [Peer Review File · Nature Communications]

Crystal structures of MHC class I complexes reveal the elusive intermediate conformations explored during peptide editingREVIEWERS' COMMENTS

Reviewer #1 (Remarks to the Author):

In an extension of work from the authors (Li et al. JBC 2019), Li et al now describe the structures of 20-mer precursor peptides loaded within HLA-B*08. These were stably entrapped using an artificial E76C mutation within HLA-B*08 coupled to a C-terminal cysteine within the "antigen" peptide termini, thus creating a stably anchored and engineered pHLA. Very high resolution structures were obtained for HIV gag-like peptides which either had a HLA-B8 Ala or Val P-2 anchor attached to a 12 amino acid N-terminal arginine-alanine extension. Either a canonical B*08 P-1 Phe was used or Ala substitution. This allowed identification of a previously unidentified region of flexibility within a portion of the MHC platform. Molecular dynamics simulations were used to try to characterise further the distortions to the peptide binding groove of the molecule. The authors conclude that significant peptide-induced structural distortions are able to occur within the HLA platform, with the nature of the antigen P1 and P2 residues strongly influencing the nearby structures adopted.

The structures are solved well, and analysed in great depth, and adds to the growing catalogue of MHC-I structures that present atypical peptides (e.g. greater than 12 amino acids), which can either bulge from the groove, or exhibit N-or C-terminal overhangs. The main weakness, if not flaw, of the paper however is the central claim surrounding whether these conformational states observed in the engineered P-HLA-B8 structures represent physiological states observed in the loading of peptides to the MHC molecules. Compounded by the fact that they do not actually observe the conformation of the loop in question (it is disordered), there are no experiments to show that the conformational states are physiologically relevant, and this is what the manuscript is really in need of. As it stands, the manuscript is of interest to MHC aficionados, but the general interest is limited.

Reviewer #2 (Remarks to the Author):

The high-resolution x-ray crystal structures of Li and coworkers visualize the "molecular gymnastics" that the peptide undergoes in the MHC I binding groove, and the coupled structural response of the MHC I molecule itself. The MHC I allele HLA-B8 with the E76C mutation was used, as the introduced Cys at position 76 enables the covalent attachment of the peptide at the C-2 position through an S-S bridge. The present work builds on previous structural biology work from the Bouvier lab (Refs 28, 29), which showed the binding mode of the (RA)₆AAKKKYCL 20mer peptide that in the present work was mutated at position 1 from Ala to Phe (FA) or, in addition, at position 2 from Ala to Val (FV). However, compared to the previous x-ray structures, the current work succeeded in visualizing the very pronounced conformational distortions of the MHC I peptide-binding groove. These results are very remarkable and valuable because they have the potential to induce a paradigm shift in the field, which so far relied exclusively on the conventional, static picture provided by the high-resolution structures available so far, and largely underestimated the important functional role of molecular dynamics and conformational ensembles ("plasticity"). Furthermore, the structures reported in the present work provide -- the up to now lacking! -- direct experimental evidence for MD simulations, especially concerning the "remarkable conformational flexibility and structural plasticity" (l. 162) of the N-terminal part of the alpha1 helix (see, for example, the increased dynamic fluctuations of the alpha1 N-term in Fig. 5B of Ref. 54, or the N-terminal groove opening seen in metadynamics MD simulations, Figs. 1D and also 10D in 10.1021/acs.jctc.9b01150).

GENERAL QUESTIONS / COMMENTS:

INTRODUCTION:

1) It would be useful for the reader to learn a bit more about the motivation as to why the authors decided to modify the extended peptides? Is it established that antigenic peptides are N-terminally cut after binding to MHC I? What would be the biological relevance of choosing 20mer peptides if ERAP can't trim MHC I-bound peptides?

2) What are the anchor residues of the peptide? It can be residue two or five, and from the introduction it's not clear to us how drastically the mutation at position two is expected to change peptide affinity. This is particularly important because the C-terminus of the peptide is bound covalently to the binding groove through an S-S bond.

RESULTS:

3) The peptide is covalently bound to the MHC I binding groove, and the crystal forms an environment that can be suspected to force every peptide N-terminus into the A-pocket region (the crystal is packed so tightly that the peptide N-terminus is just squeezed in -- especially the bulky extensions might form a kind of cantilever via which crystal packing can influence the binding of the first two residues of the peptide.) Why are the structural changes observed at the A-pocket region relevant in solution/in the cell? Isn't there a risk that the crystal design just squeezed the peptides into the binding groove, even "accepting" to deform the A-pocket region in a way that would never happen inside a cell?

4) Residues 54-59 are not resolved in the x-ray structures with the 20mer "FA" and "FV" peptides bound, so this part of the protein structure needed to be modelled to generate a starting structure for the MD simulations. Then, in the MD, exactly these modelled residues are the most flexible ones. How can the authors exclude that this is somehow linked to the modelling in the first step? For example, is it conceivable that the initial conformation generated by the modelling was not optimal, and therefore needed to relax during the simulations, which would lead to a "slow structural drift" as a fctn of simulation time? To address this potential issue, the authors should discard longer and longer initial parts of their simulation trajectories (as "equilibration/relaxation time") and check whether the results stay consistent. The simulations might need to be extended to do that reliably, because the equilibration times are rather short and the sampling is rather limited, questioning to what extent the MD simulations can yield reliable equilibrium distributions.

DETAILED COMMENTS:

5) The scheme in Fig. 7 is highly speculative. It is not supported by any data and should therefore be removed. As far as we can see, the authors have no data on the free energy landscape, such as the Boltzmann populations of the minima and the barriers (and their heights). Can the authors come up with a different, more schematic Figure to illustrate their point?

6) In line 68, the authors speak of "non-native" conformations. We guess that what they mean here is that the conformation differs from the known "canonical" MHC I structure. This might be called "non-native" -- however, it could be explained (also at other positions in the text) that i) this is simply a conformationally excited state (or an ensemble of such states), which is slightly higher in (free) energy but thermally accessible (in other words, has non-zero Boltzmann population at physiological T), and ii) which plays an essential role for function (the latter message is nicely laid out by the authors, the former (more biophysics perspective) not so much in our opinion).

7) Somewhat along the same lines as in point 1 above: In lines 424 and 862 (Fig. 2 caption), the authors speak of "correctly conformed molecules" or a "correctly conformed groove", a wording that somewhat implies that there is sth "incorrect" with the alternative conformations observed. However, that is not the case. The present work is exactly a very nice example that directly shows -- at Angstrom scale structural resolution -- that proteins are dynamic, and that proteins can better be understood in terms of (more or less broad) conformational ensembles instead of as single static snapshots. The present work is a nice example for how also x-ray crystallography can capture these natural dynamics.

8) In lines 239 and 252, the authors interpret their MD simulations in terms of the "stability" of certain interactions (P1-Tyr59 and P1-Ile52). From broader distance distributions, they infer that these interactions are "less stable". We disagree with this interpretation. A broader distributions simply means that the minimum in the underlying energy landscape is broader/shallower. No conclusions on the "stability" can be drawn from that, at least not in such a simple way. Furthermore, it is not specified what the authors mean by "stability", structural or thermodynamic/energetic (or both)?

9) Line 248, 257 / Fig. 5: How exactly is the "inter-residue distance" defined? What C-H-X angle (what C-H bond exactly)? We see that this is probably explained in Methods, but it would be helpful to add this to the Figure caption text.

10) Line 282: Of course, different repeats of the MD simulations give different "evolution curves". Molecular dynamics are stochastic in nature, and in this sense the pathways are never "unique". So this is not worth mentioning and should be rephrased. Furthermore, we suggest to write "time evolution" or "time traces" or so instead of "evolution curves" (also elsewhere in the text, e.g., l. 553, Fig. S6).

11) Line 556: The 10 ns running average has to be mentioned in Fig. S6 caption text. In addition, the actual raw data should be plotted as well, maybe with the running averages "on top" as thick lines.

12) Line 162: What is, in the authors view, the difference between "conformational flexibility" and "structural plasticity"? If these are used as synonyms, one of the two suffices (and reduces possible misunderstandings).

MINOR ISSUES:

In line 515, it should not be "constraints" but "harmonic position restraints". Constraints are used for bond lengths, see line 509.

Line 516: "The time interval for conformational sampling was 20 ps." sounds a little odd. "Coordinates were saved to disk every 20 ps." might be better?

Line 521: "attempt swap duration of 1 ps" should be changed to "swap attempt frequency of 1/ps".

Line 535: Typo: reproducibility

Reviewer #3 (Remarks to the Author):

This body of work presents four new crystal structures of MHC-I molecules, two of which exhibit a partially empty peptide binding groove. The authors leverage a human MHC-I molecule with an additional P5 peptide anchor and peptide-bridging disulfide bond (HLA-B*08E76C), to obtain X-ray structures of N-terminally extended peptides, relative to their 8mer counterparts, which exhibit missing density at the A, B pockets (likely due to decrease in ordered regions and/or increased protein conformational heterogeneity/dynamics), and reveal the presence of conformational adaptations. These findings are complemented by molecular dynamics simulations which suggest similar conformational adaptations and a plausible structural mechanism. Based on these results, the authors hypothesize that their structures correspond to transient substates explored by MHC I molecules during the peptide loading and editing process.

Elucidating structural intermediates along the MHC-I loading pathway is an important, outstanding question with relevant implications for the design of peptide vaccines and for understanding the emergence of immunodominant antigens, and therefore the work is highly significant to the broader molecular immunology community. Moreover, while snapshots of MHC-I molecules with missing peptide occupancy of the F-pocket have been characterized in several studies, important details about how peptides bind to the A,B pockets remain incompletely characterized. This work uses an elegant system to obtain insights that highlight the importance of specific interactions with MHC-I residues (IY59, I52), which stabilize the peptide N-terminus (in an analogous manner to the role of dynamics of the alpha2-1 helix for stabilizing the peptide C-terminus). Overall, the data are of high quality and the work is presented in a clear and succinct manner. It also poses the intriguing and novel hypothesis that such "mobile" elements near the A, B pockets may create

conformational epitopes for yet to be identified molecular chaperones. However, the hypothesis that the observed crystallographic snapshots correspond to naturally occurring intermediates along the peptide loading pathway (if a well-defined pathway exists), is not tested further in the present work, and therefore the relevance of these structural findings for the MHC-I peptide loading pathway is uncertain. In addition, there are important technical limitations of the study, outlined in detail below:

1) While the combined crystallographic and MD analysis suggest the presence of significant conformational dynamics (or so-called "motions" in the manuscript) for specific MHC-I regions, this is yet to be validated in a solution environment. Moreover, whether the missing density arises from functionally relevant protein motions, remains to be conclusively addressed. The authors can complement their structural data with some measurement of solution dynamics at ambient temperatures (such as hydrogen/deuterium exchange mass spectroscopy), to further complement and validate their crystallographic results, obtained under cryogenic conditions.

2) The authors employ an artificial system, where the peptide is tethered to the MHC-I groove via an engineered disulfide bond at P7. While this is essential to prepare complexes with longer peptides, it also raises questions about the physiological relevance of these results. How does the presence of the disulfide expected to affect the conclusions drawn in the manuscript regarding significant structural adaptations in the A, B pockets and loss of density in the MHC-I 310 helix? This should be addressed in a revised version of the manuscript by a suitable experimental or computational technique.

3) The timescale of the MD simulations may be too short to capture the conformational adaptations that are hypothesized by the authors. In order to better support their conclusions, the authors should consider extending their simulations to the microsecond timescale, or employ enhanced sampling protocols.

4) Ln 233-234, 270-271: time traces of the inter-residue distance between P1 and Tyr59, the P1 omega angle, and any other structural parameters extracted from the simulations should be provided, to examine whether the simulations have established ergodicity with respect to these parameters, and that different free energy barriers involved can be transversed reversibly. Could it be that the simulation results arise from irreversible conformational adaptations (or local optimization) of the refined model coordinates to a new simulation force field?

5) How consistent are these results in simulations performed using different force fields?

6) Ln. 192 cites similar Tm values for the 20mer peptide systems, however it is more likely that the presence of the artificial disulfide overshadows any difference in peptide binding free energies, as suggested by the observed structural differences and network of interactions. More specific thermodynamic parameters such as binding enthalpies and entropies from ITC are needed to draw any conclusions about the effect of these interactions on peptide binding thermodynamics, and to establish the presences of intermediate states.

Other comments:

Ln 142-143: "To the best of our knowledge, this is the first report of MHC I structures, with or without a bound peptide, showing such significant conformational distortions in the groove." This statement is inaccurate, because the structures presented in this work all carry a bound peptide, and because MHC-I structures with various peptide occupancies have been published previously (e.g. Anjanappa et al., Nat commun, 2020; Hafstrand et al., PNAS, 2019)

Ln 207-208: "data not shown" structural overlays of the C-F pockets should be provided to support this claim.

The final section of Results "Analysis of our structures in the context of bat MHC I molecules and human MHC II molecules" appears more speculative / suggestive in nature and is not supported by any further results. This section should be moved into the discussion.

An analysis of any structured water molecules between the different models can provide additional insights into the role of solvation for the observed conformational adaptations in the A, B pockets.

We thank all Reviewers for their comments aimed at improving our manuscript. We appreciate their time and efforts. We have responded below.

Reviewer #1 (Remarks to the Author):

In an extension of work from the authors (Li et al. JBC 2019), Li et al now describe the structures of 20-mer precursor peptides loaded within HLA-B*08. These were stably entrapped using an artificial E76C mutation within HLA-B*08 coupled to a C-terminal cysteine within the “antigen” peptide termini, thus creating a stably anchored and engineered pHLA. Very high resolution structures were obtained for HIV gag-like peptides which either had a HLA-B8 Ala or Val P-2 anchor attached to a 12 amino acid N-terminal arginine-alanine extension. Either a canonical B*08 P-1 Phe was used or Ala substitution. This allowed identification of a previously unidentified region of flexibility within a portion of the MHC platform. Molecular dynamics simulations were used to try to characterise further the distortions to the peptide binding groove of the molecule. The authors conclude that significant peptide-induced structural distortions are able to occur within the HLA platform, with the nature of the antigen P1 and P2 residues strongly influencing the nearby structures adopted.

The structures are solved well, and analysed in great depth, and adds to the growing catalogue of MHC-I structures that present atypical peptides (e.g. greater than 12 amino acids), which can either bulge from the groove, or exhibit N-or C-terminal overhangs. The main weakness, if not flaw, of the paper however is the central claim surrounding whether these conformational states observed in the engineered P-HLA-B8 structures represent physiological states (physiological-like states) observed in the loading of peptides to the MHC molecules. Compounded by the fact that they do not actually observe the conformation of the loop in question (it is disordered), there are no experiments to show that the conformational states are physiologically relevant, and this is what the manuscript is really in need of. As it stands, the manuscript is of interest to MHC aficionados, but the general interest is limited.

Answer: For more than 10 years, independent groups have consistently reported on the molecular dynamics properties of MHC I molecules, and have highlighted how regions of dynamics in MHC I form recognition hot spots for accessory proteins and how these interaction support high-affinity peptide selection (Refs. 7-15). It is therefore increasingly appreciated that MHC I is more than a static peptide-presenting platform, and that molecular motions in MHC I are modulated by peptides. However, no one has ever shown crystallographically (or by other techniques) how molecular dynamics “looks like” in MHC I, how do MHC I-peptide intermediates “look like”, and how peptide-dependent conformational fluctuations can shape MHC I molecules. Our current results provide the first examples of unusual and intriguing molecular features in MHC I and MHC I-peptide interaction in support of the notion that 1. MHC I is a dynamic protein; and 2. that peptide serve as modulating agents of MHC I motional properties. That we do not observe “the loop in question” adds even more credibility to MHC I undergoing pronounced peptide-dependent molecular motions. Our results provide direct evidence of molecular dynamics in MHC I, and insights into how this inherent property can support MHC I peptide-selector function (Abstract; Lines 77 to 82; and Conclusion). Our results should not be viewed as only “adding to the growing catalogue of MHC I structures...”.

More than 90% of the studies that have characterized dynamics in MHC I as it pertains to its peptide-selector function have focused on the C-terminus of the groove. This is because there is more information published on accessory proteins that recognize conformational hot spots at the C-terminus. This knowledge is lacking for the N-terminus of the groove (see Line 60). Hence, addressing the question of this Reviewer is extremely challenging for now. Our manuscript plants the first seed to tackle this question; we need to identify which proteins are associated with maturation and/or peptide-editing at the N-terminus of the groove and incorporate them in the system – see Discussion, start Line 410. Our current work is at the forefront of efforts to be made towards this

goal. It is worth noting that only a very small number of studies have addressed the functional relevance of motions at the C-terminus of the groove, in spite of all of the work done in the last 10 years: “*Peptide exchange on MHC-I by TAPBR is driven by a negative allostery release cycle*” ([10.1038/s41589-018-0096-2](https://doi.org/10.1038/s41589-018-0096-2)); “*Molecular determinants of chaperone interactions on MHC-I for folding and antigen repertoire selection*” (<https://doi.org/10.1073/pnas.1915562116>); and “*Structural mechanism of tapasin-mediated MHC I-peptide loading in antigen presentation*” ([10.1038/s41467-022-33153-8](https://doi.org/10.1038/s41467-022-33153-8)).

On line 406, we say clearly that the exact molecular features of MHC I-peptide intermediates are expected to be fluid; our structures are examples of what is possible molecularly when MHC I “stretches” itself in response to bound candidate peptides. We have rephrased some sentences in the manuscript to eliminate wordings that could imply our structures are *exactly* as seen inside cells (it was in the Abstract and Introduction).

See also Response 1) of Reviewer 3.

Reviewer #2 (Remarks to the Author):

The high-resolution x-ray crystal structures of Li and coworkers visualize the “molecular gymnastics” that the peptide undergoes in the MHC I binding groove, and the coupled structural response of the MHC I molecule itself. The MHC I allele HLA-B8 with the E76C mutation was used, as the introduced Cys at position 76 enables the covalent attachment of the peptide at the C-2 position through an S-S bridge. The present work builds on previous structural biology work from the Bouvier lab (Refs 28, 29), which showed the binding mode of the (RA)₆AAKKKYCL 20mer peptide that in the present work was mutated at position 1 from Ala to Phe (FA) or, in addition, at position 2 from Ala to Val (FV). However, compared to the previous x-ray structures, the current work succeeded in visualizing the very pronounced conformational distortions of the MHC I peptide-binding groove. These results are very remarkable and valuable because they have the potential to induce a paradigm shift in the field, which so far relied exclusively on the conventional, static picture provided by the high-resolution structures available so far, and largely underestimated the important functional role of molecular dynamics and conformational ensembles (“plasticity”). Furthermore, the structures reported in the present work provide -- the up to now lacking! -- direct experimental evidence for MD simulations, especially concerning the “remarkable conformational flexibility and structural plasticity” (l. 162) of the N-terminal part of the alpha1 helix (see, for example, the increased dynamic fluctuations of the alpha1 N-term in Fig. 5B of Ref. 54, or the N-terminal groove opening seen in metadynamics MD simulations, Figs. 1D and also 10D in [10.1021/acs.jctc.9b01150](https://doi.org/10.1021/acs.jctc.9b01150)).

GENERAL QUESTIONS / COMMENTS:

INTRODUCTION:

1) It would be useful for the reader to learn a bit more about the motivation as to why the authors decided to modify the extended peptides? Is it established that antigenic peptides are N-terminally cut after binding to MHC I? What would be the biological relevance of choosing 20mer peptides if ERAP can't trim MHC I-bound peptides?

Answer: About our motivation: In the Introduction (start Line 62), we have elaborated more on why we have undertaken this work. Basically, we want to link together N-terminal region of MHC I groove/peptide-dependent conformations adjustments in MHC I /MHC I conformational flexibility/functional relevance for peptide editing/role of ERAPs. Although significant efforts have been spent to integrate these concepts for the C-terminus of the groove (minus ERAPs), much less is known about the role of the N-terminus of the groove in MHC I peptide-selector function (see Line 60 and Response to Reviewer 1). Hence, we have a fragmented understanding of the MHC I

groove, when a global understanding of the groove is needed. In the Discussion (start Line 410), we elaborate on the functional implication of our results for MHC I maturation and/or peptide editing including the role of ERAPs in trimming MHC I-bound peptides.

About ERAPs: The role of ERAP in trimming MHC I-bound peptides is not as well established as trimming of free peptides but a number of studies support this function as discussed starting on Line 422. Our current results add one more piece of supportive evidence and raise an interesting hypothesis on the N-terminus's role in peptide editing.

About 20mer peptides: We showed crystallographically (Ref. 30) for (RA)_nAAKKKYCL that 10, 12, 14, and 20mers (n = 2 - 6) bind identically in the groove with their (RA)_n residues overhanging at the N-terminus and their P1 side chains rotating in the A pocket (Lines 65-68) – any of these peptides would therefore have generated a “trapped” conformation after substitutions at P1 and P2. We choose a 20mer peptide because we showed previously that peptides must have at least 5-6 residues extending out of the groove (Ref. 30), i.e., at least 14mer in length, to be sensitive to ERAP trimming while bound to MHC I. We added a peptide length specificity in the Discussion (Line 424).

2) What are the anchor residues of the peptide? It can be residue two or five, and from the introduction it's not clear to us how drastically the mutation at position two is expected to change peptide affinity. This is particularly important because the C-terminus of the peptide is bound covalently to the binding groove through an S-S bond.

Answer: Based on deposited x-ray structures of HLA-B8/peptide complexes, HLA-B8 has a middle anchor residue at P5 and a C-terminal anchor at P9 (or P8), and it can also have a N-terminal anchor residue at either P2 or P3 (depending on the peptide sequence) – note that mass spectrometry analyses of eluted peptides usually assign P3 as the N-terminal anchor but this is not be entirely correct based on X-ray structures. In our structures, the P2 and P3 side chains point side way (see Supplementary Figure 1), even in the FA and FV 8mer structures, so both side chains serve as “moderate” N-terminal anchors. Consequently, P2 substitutions in FA and FV 8mer did not affect the thermostabilities: T_m is 71.1°C for FA and 71.4°C for FV (Line 197). A similar conclusion can be made for FA and FV 20mer: 65.8°C FA and 67.5°C FV 20mer (Line 194). Importantly, in our previous work with HLA-B8 (ref. 32), we measured identical T_m for AAKKKYCL (disulfide-bonded to HLA-B8E76C) and AAKKKYKL (not disulfide-bonded to HLA-B8), indicating that the S-S bond does not alter thermostability (and the peptides were bound identically in the groove based on their x-ray structures). See also Response 3) below.

RESULTS:

3) The peptide is covalently bound to the MHC I binding groove, and the crystal forms an environment that can be suspected to force every peptide N-terminus into the A-pocket region (the crystal is packed so tightly that the peptide N-terminus is just squeezed in -- especially the bulky extensions might form a kind of cantilever via which crystal packing can influence the binding of the first two residues of the peptide.) Why are the structural changes observed at the A-pocket region relevant in solution/in the cell? Isn't there a risk that the crystal design just squeezed the peptides into the binding groove, even "accepting" to deform the A-pocket region in a way that would never happen inside a cell?

Answer: To address potential issues on peptides being disulfide bonded to the MHC I groove, we repeated our analyses of FA 20mers and FV 20mers with this disulfide bond reduced, i.e., MD simulations and RMSF analysis of MHC I heavy chain, for P1-Tyr59 and P1-Ile52 interactions, as well as P1 rotation in the A pocket (Supplementary Figs. 6 and 8, respectively). The results show that reducing the disulfide bond had minimal effects on the behavior of the bound 20mers. Thus, the structural adaptation in pockets A and B (Supplementary Fig. 6) and P1 rotation (Supplementary Fig. 8) for engineered disulfide bond at P7 is validated. This is explained in Methods (Line 504). New text was also added starting on Lines 270 and 293.

Why are the structural changes observed at the A-pocket region relevant in solution/in the cell?
Please see Response to Reviewer 1.

4) Residues 54-59 are not resolved in the x-ray structures with the 20mer "FA" and "FV" peptides bound, so this part of the protein structure needed to be modelled to generate a starting structure for the MD simulations. Then, in the MD, exactly these modelled residues are the most flexible ones. How can the authors exclude that this is somehow linked to the modelling in the first step? For example, is it conceivable that the initial conformation generated by the modelling was not optimal, and therefore needed to relax during the simulations, which would lead to a "slow structural drift" as a fctn of simulation time? To address this potential issue, the authors should discard longer and longer initial parts of their simulation trajectories (as "equilibration/relaxation time") and check whether the results stay consistent. The simulations might need to be extended to do that reliably, because the equilibration times are rather short and the sampling is rather limited, questioning to what extent the MD simulations can yield reliable equilibrium distributions.

Answer: We thank the reviewer for this suggestion. For FA and FV 20mers, we have extended each REMD simulation time to 90 ns. In addition, we also extended other systems with complete pdb structures (AA20 and control 8mers) to 50 ns. We have updated this information in Methods (Line 532). Based on the suggestions, we examined equilibration of our MHC I systems. The figure below shows the RMSF curves obtained from the ensembles from different cut-off of the initial parts of the simulation trajectory, where "All" means that the whole trajectory was used, while "All-xxns" means that the RMSF was calculated from the ensemble with the first "xxns" of the trajectory discarded. The results show that the RMSF curves did not change with different cut-offs of the initial parts of the simulation trajectories, indicating that the systems are well equilibrated. In the manuscript, we have updated our panels in Figs. 5, S5, and S7 (was S6) using results of the extended simulations. We hope that this analysis will remove all concerns. See also Response 3) of Reviewer 3.

DETAILED COMMENTS:

5) The scheme in Fig. 7 is highly speculative. It is not supported by any data and should therefore be removed. As far as we can see, the authors have no data on the free energy landscape, such as the

Boltzmann populations of the minima and the barriers (and their heights). Can the authors come up with a different, more schematic Figure to illustrate their point?

Answer: We removed Fig. 7 and rephrased Lines 345 and 397 in the Discussion.

6) In line 68, the authors speak of "non-native" conformations. We guess that what they mean here is that the conformation differs from the known "canonical" MHC I structure. This might be called "non-native" -- however, it could be explained (also at other positions in the text) that i) this is simply a conformationally excited state (or an ensemble of such states), which is slightly higher in (free) energy but thermally accessible (in other words, has non-zero Boltzmann population at physiological T), and ii) which plays an essential role for function (the latter message is nicely laid out by the authors, the former (more biophysics perspective) not so much in our opinion).

Answer: We removed "native" and "non-native" from the manuscript since we agree that all structures in our current work can be viewed as "native". We now use "canonical" to specifically refer to the "final" MHC I structures, i.e., those MHC I structures with standard features that have been published for the last 30 years. We use "canonical" for structure/position/conformation/side chain/fold/etc. We do not use "non-canonical".

7) Somewhat along the same lines as in point 1 above: In lines 424 and 862 (Fig. 2 caption), the authors speak of "correctly conformed molecules" or a "correctly conformed groove", a wording that somewhat implies that there is sth "incorrect" with the alternative conformations observed. However, that is not the case. The present work is exactly a very nice example that directly shows -- at Angstrom scale structural resolution -- that proteins are dynamic, and that proteins can better be understood in terms of (more or less broad) conformational ensembles instead of as single static snapshots. The present work is a nice example for how also x-ray crystallography can capture these natural dynamics.

Answer: We see the point of the Reviewer. We removed from the manuscript "correctly conformed molecules" and "correctly conformed groove".

8) In lines 239 and 252, the authors interpret their MD simulations in terms of the "stability" of certain interactions (P1-Tyr59 and P1-Ile52). From broader distance distributions, they infer that these interactions are "less stable". We disagree with this interpretation. A broader distribution simply means that the minimum in the underlying energy landscape is broader/shallower. No conclusions on the "stability" can be drawn from that, at least not in such a simple way. Furthermore, it is not specified what the authors mean by "stability", structural or thermodynamic/energetic (or both)?

Answer: We rephrase the interpretation of the distribution of inter-residue distances (Lines 246 and 261)

9) Line 248, 257 / Fig. 5: How exactly is the "inter-residue distance" defined? What C-H-X angle (what C-H bond exactly)? We see that this is probably explained in Methods, but it would be helpful to add this to the Figure caption text.

Answer: We added the definitions of the "inter-residue distance" and "C-H-X angle" in the legends of Fig. 5 and Supplementary Fig. 5B and Figs. 6B, C, and E.

10) Line 282: Of course, different repeats of the MD simulations give different "evolution curves". Molecular dynamics are stochastic in nature, and in this sense the pathways are never "unique". So this is not worth mentioning and should be rephrased. Furthermore, we suggest to write "time evolution" or "time traces" or so instead of "evolution curves" (also elsewhere in the text, e.g., l. 553, Fig. S6).

Answer: We eliminated the sentence on unique pathways. We have also used the suggested term "time traces" -- Lines 287, 560, 562, 565, and Supplementary Figs. 7B,C and 8A,B.

11) Line 556: The 10 ns running average has to be mentioned in Fig. S6 caption text. In addition, the actual raw data should be plotted as well, maybe with the running averages "on top" as thick lines.
Answer: As requested, the "10 ns running average" was added in the legend of Supplementary Figs. 7B and C (was Fig. S6B and C) and we now show the running averages as black lines. Same for data in Supplementary Fig. 8.

12) Line 162: What is, in the authors view, the difference between "conformational flexibility" and "structural plasticity"? If these are used as synonyms, one of the two suffices (and reduces possible misunderstandings).

Answer: We used these two terms interchangeably in the text. We have now removed "plasticity" from the manuscript except on Line 144.

MINOR ISSUES:

In line 515, it should not be "constraints" but "harmonic position restraints". Constraints are used for bond lengths, see line 509.

Answer: Changed as suggested, Line 522.

Line 516: "The time interval for conformational sampling was 20 ps." sounds a little odd. "Coordinates were saved to disk every 20 ps." might be better?

Answer: Changed as suggested, Line 524.

Line 521: "attempt swap duration of 1 ps" should be changed to "swap attempt frequency of 1/ps".

Answer: Changed as suggested, Line 529.

Line 535: Typo: reproducibility

Answer: Fixed, Line 544.

Reviewer #3 (Remarks to the Author)

This body of work presents four new crystal structures of MHC-I molecules, two of which exhibit a partially empty peptide binding groove. The authors leverage a human MHC-I molecule with an additional P5 peptide anchor and peptide-bridging disulfide bond (HLA-B*08E76C), to obtain X-ray structures of N-terminally extended peptides, relative to their 8mer counterparts, which exhibit missing density at the A, B pockets (likely due to decrease in ordered regions and/or increased protein conformational heterogeneity/dynamics), and reveal the presence of conformational adaptations. These findings are complemented by molecular dynamics simulations which suggest similar conformational adaptations and a plausible structural mechanism. Based on these results, the authors hypothesize that their structures correspond to transient substates explored by MHC I molecules during the peptide loading and editing process.

Elucidating structural intermediates along the MHC-I loading pathway is an important, outstanding question with relevant implications for the design of peptide vaccines and for understanding the emergence of immunodominant antigens, and therefore the work is highly significant to the broader molecular immunology community. Moreover, while snapshots of MHC-I molecules with missing peptide occupancy of the F-pocket have been characterized in several studies, important details about how peptides bind to the A,B pockets remain incompletely characterized. This work uses an elegant system to obtain insights that highlight the importance of specific interactions with MHC-I residues (IY59, I52), which stabilize the peptide N-terminus (in an analogous manner to the role of

dynamics of the alpha2-1 helix for stabilizing the peptide C-terminus). Overall, the data are of high quality and the work is presented in a clear and succinct manner. It also poses the intriguing and novel hypothesis that such “mobile” elements near the A, B pockets may create conformational epitopes for yet to be identified molecular chaperones. However, the hypothesis that the observed crystallographic snapshots correspond to naturally occurring intermediates along the peptide loading pathway (if a well-defined pathway exists), is not tested further in the present work, and therefore the relevance of these structural findings for the MHC-I peptide loading pathway is uncertain. In addition, there are important technical limitations of the study, outlined in detail below:

1) While the combined crystallographic and MD analysis suggest the presence of significant conformational dynamics (or so-called “motions” in the manuscript) for specific MHC-I regions, this is yet to be validated in a solution environment. Moreover, whether the missing density arises from functionally relevant protein motions, remains to be conclusively addressed. The authors can complement their structural data with some measurement of solution dynamics at ambient temperatures (such as hydrogen/deuterium exchange mass spectroscopy), to further complement and validate their crystallographic results, obtained under cryogenic conditions.

Answer: The question of whether the observed motions are functionally relevant will be progressively answered as we gain more knowledge of how the N-terminus of the groove influences peptide editing (see Response to Reviewer 1). NMR is somewhat ideal for monitoring protein motions, but as mentioned in our response to Reviewer 1, only a handful of papers in more than 10 years have addressed the question raised by this Reviewer for motions at the C-terminus of the groove. These NMR studies have, as expected, shown that conformational changes in MHC I observed crystallographically reflect the propensity of MHC I to move.

2) The authors employ an artificial system, where the peptide is tethered to the MHC-I groove via an engineered disulfide bond at P7. While this is essential to prepare complexes with longer peptides, it also raises questions about the physiological relevance of these results. How does the presence of the disulfide expected to affect the conclusions drawn in the manuscript regarding significant structural adaptations in the A, B pockets and loss of density in the MHC-I 310 helix? This should be addressed in a revised version of the manuscript by a suitable experimental or computational technique.

Answer: See Response 3) of Reviewer 2.

3) The timescale of the MD simulations may be too short to capture the conformational adaptations that are hypothesized by the authors. In order to better support their conclusions, the authors should consider extending their simulations to the microsecond timescale, or employ enhanced sampling protocols.

Answer: We have now extended the timescale of all plain MD simulation for rotation of P1 peptide N-terminal amino group to 1 μ s. The new simulations are shown in Supplementary Figs. 7B and C (and S8 for non-disulfide bonded complexes). This extension to microsecond timescale did not change our main conclusion on the ability of the terminal amino group to rotate in the A pocket. For REMD simulations on the flexibility of the N-terminal of MHC I residues and the interaction of P1-Y59 and P1-I52, we also extend the simulations of FA and FV 20mers to 90 ns for each replica. The new results are shown in Fig. 5 and the convergence checks were shown above (see Response 4) of Reviewer 2). Extension of MD simulations does not change our main conclusions.

4) Ln 233-234, 270-271: time traces of the inter-residue distance between P1 and Tyr59, the P1 omega angle, and any other structural parameters extracted from the simulations should be provided, to examine whether the simulations have established ergodicity with respect to these

parameters, and that different free energy barriers involved can be transversed reversibly. Could it be that the simulation results arise from irreversible conformational adaptations (or local optimization) of the refined model coordinates to a new simulation force field?

Answer: Based on the reviewer's suggestion, we drew the time trace of inter-residue distances and angles in Supplementary Fig. 5 for the replica MD simulation at 310K. The figure below shows that all structural parameters (including the distances of P1...Y59 and P1...I52, the relative orientation between P1 and Y59, and C-H-X angle in Supplementary Fig. 5B) fluctuate back and forth in a wide range, indicating that the simulations have established ergodicity and reached thermal equilibration. We hope that this analysis will remove all concerns.

5) How consistent are these results in simulations performed using different force fields?

Answer: We thank the reviewer for the comment. In our current work, we employed CHARMM36m force field in MD simulations, which is one of the most widely used all-atom force field and it has been used previously in simulating other MHC I systems, see as examples [D.R. Bell *et al.* 2021] *J. Chem. Theory Comput.* 17, 7962–7971; T. Arns *et al.* (2020) *Front. Immunol.* 11:575076; L. Susac *et al.* (2022) *Cell* 185, 3201–3213]. Moreover, our simulation results are entirely consistent with our experimental results (x-ray structures) indicating that the current force field is proper enough for simulating the MHC I system. We could use other forcefields such as Amber99SB for MD simulations and benchmark the force fields with our experimental results, but this is in our opinion outside the scope of our current manuscript.

6) In. 192 cites similar T_m values for the 20mer peptide systems, however it is more likely that the presence of the artificial disulfide overshadows any difference in peptide binding free energies, as suggested by the observed structural differences and network of interactions. More specific thermodynamic parameters such as binding enthalpies and entropies from ITC are needed to draw any conclusions about the effect of these interactions on peptide binding thermodynamics, and to establish the presences of intermediate states.

Answer: As mentioned by Reviewer 2 comment 7), our structures nicely exemplify that native structures of MHC I are dynamic and made up of an ensemble of structures. We have captured these natural dynamics in way that has never been done before. See also Responses 6) and 7) of Reviewer 2.

Other comments:

Ln 142-143: “To the best of our knowledge, this is the first report of MHC I structures, with or without a bound peptide, showing such significant conformational distortions in the groove.”

This statement is inaccurate, because the structures presented in this work all carry a bound peptide, and because MHC-I structures with various peptide occupancies have been published previously (e.g. Anjanappa et al., Nat commun, 2020; Hafstrand et al., PNAS, 2019)

Answer: The two reference provided by the Reviewer did reveal minor changes in side chain configurations and no changes in MHC I structure. Others have made similar statements on minor conformational changes seen in MHC I thus far – see for example “*Dynamically driven allostery in MHC proteins: peptide-dependent tuning of class I MHC global flexibility*” (<https://doi.org/10.3389/fimmu.2019.00966>), first paragraph of the Discussion. We have nonetheless rephrased the sentence as “To the best of our knowledge, this is the first report of an MHC I structure that shows pronounced distortions in the groove.” (Line 144).

Ln 207-208: “data not shown” structural overlays of the C-F pockets should be provided to support this claim.

Answer: We cannot present 12 panels (FA 8/20 and FV 8/20 for six pockets A-F) in one figure. However, to address the point raised, we have now added one new panel in Supplementary Figure 4 that shows no changes in MHC I side chain orientations in pocket C. We use this panel as a reference for pockets D-F on Line 216 and Legend of Fig. S4.

The final section of Results “Analysis of our structures in the context of bat MHC I molecules and human MHC II molecules” appears more speculative / suggestive in nature and is not supported by any further results. This section should be moved into the discussion.

Answer: In Figure 6, we carried out a comparative structural analysis of our structures with bat and MHC II molecules; we view this analysis as “results” and thus suited for this section. We have however shortened our interpretation of conformational flexibility in bat MHC I molecules to eliminate what could be considered speculations (Line 306).

An analysis of any structured water molecules between the different models can provide additional insights into the role of solvation for the observed conformational adaptations in the A, B pockets.

Answer: There are two structured water molecules in pocket A of FA and FV 8mer that we have now added to each panel in Fig. 4B. These water molecules are hydrogen bonded to several MHC I residues, including a direct interaction to Tyr59. Because Tyr59 is disordered in FA and FV 20mer, these water molecules are missing in the 20mer structures (Fig. 4A). We have commented on this important point in Lines 198 and 392 and Fig. 4B legend.

REVIEWERS' COMMENTS

Reviewer #2 (Remarks to the Author):

We are satisfied with the changes made. In particular, the authors did a good job in addressing this reviewer's questions and recommendations, for example by extending the MD simulations by a significant amount and also by running additional MD simulations with the disulphide bond between peptide and MHC I reduced. The results of these additional checks and validations support the previous findings.

Reviewer #3 (Remarks to the Author):

The authors have adequately addressed my comments, and provided new results that bolster the conclusions from their original work. I recommend publication of the revised manuscript in Nature communications.